# Learning Gibbs-regularized GANs with variational discriminator reparameterization

## Abstract

We propose a novel approach to regularizing generative adversarial networks (GANs) leveraging learned *structured Gibbs distributions*. Our method consists of reparameterizing the discriminator to be an explicit function of two densities: the generator PDF $q$ and a structured Gibbs distribution $\nu$. Leveraging recent work on invertible pushforward density estimators, this reparameterization is made possible by assuming the generator is invertible, which enables the analytic evaluation of the generator PDF $q$. We further propose optimizing the Jeffrey divergence, which balances mode coverage with sample quality. The combination of this loss and reparameterization allows us to effectively regularize the generator by imposing structure from domain knowledge on $\nu$, as in classical graphical models. Applying our method to a vehicle trajectory forecasting task, we observe that we are able to obtain quantitatively superior mode coverage as well as better-quality samples compared to traditional methods.

## 1 Introduction

Although recent progress in GANs and variational methods have significantly advanced the capabilities of generative models for high-dimensional data, many issues still limit the practical application of such methods. In practice, GANs typically suffer from mode loss, whereas VAEs suffer from poor sample quality compared to GANs [40, 14, 3]. Several factors may be identified as contributing to these issues: first, the training loss may optimize for sample quality at the expense of mode coverage or vice-versa; second, the variance of stochastic gradient estimates may be too high to admit efficient stochastic optimization; finally, the models may not admit a good regularization scheme via the imposition of appropriate inductive biases.

We advocate a novel approach to address these concerns, with a particular focus on the latter issue of regularization. Let $p : \mathbb{R}^N \to \mathbb{R}^+$ denote the PDF of a continuous data distribution and $q : \mathbb{R}^N \to R^+$ denote the PDF of a learned model. Following recent work [31], we advocate training a generative model to minimize the *Jeffrey* (symmetric KL) divergence $\min_q \mathrm{KL}(p, q) + \mathrm{KL}(q, p)$, representing $q$ in a way that enables it to be efficiently evaluated at any point (i.e., by representing $q$ as a *pushforward* distribution induced by an invertible warp [29, 8, 18]). In this work, we propose the key innovation of applying Fenchel-duality-based variational inference to $\mathrm{KL}(q, p)$, which allows the latter to be optimized without having to explicitly evaluate $p$. This yields the following variational approximation of the Jeffrey divergence, which constitutes the training loss for our method:

$$\min_{q \in \mathcal{Q}} \mathbb{E}_{\hat{x} \sim p} \log \frac{p(\hat{x})}{q(\hat{x})} + \sup_{\nu > 0} -\mathbb{E}_{\hat{x} \sim p} \frac{q(\hat{x})}{\nu(\hat{x})} + \mathbb{E}_{x \sim q} \log \frac{q(x)}{\nu(x)}, \qquad (1)$$

where the variational parameters consist of the function $\nu$. We now assert and later elaborate the following properties of the Jeffrey divergence and its variational approximation (1): first, $\mathrm{KL}(p, q)$ essentially prevents mode loss, whereas $\mathrm{KL}(q, p)$ prevents the generation of any samples unlikely under the data [6, 14]; second, an exactly unbiased stochastic gradient of $\mathrm{KL}(p, q)$ may be obtained without variational inference; finally, the optimal value of $\nu$ is $p$. The second key innovation of our work is to recognize that—in contrast to traditional GANs, which admit no comparable regularization principle—this model may be effectively regularized by imposing any domain-specific structure possessed by $p$ on $\nu$, which we thence interpret as a *structured Gibbs distribution*.

These concepts are illustrated in Fig. 1. Figure 1a shows the result of training a model $q$ to optimize $\mathrm{KL}(p, q)$, where $q$ is represented as the pushforward of a Gaussian base distribution under an

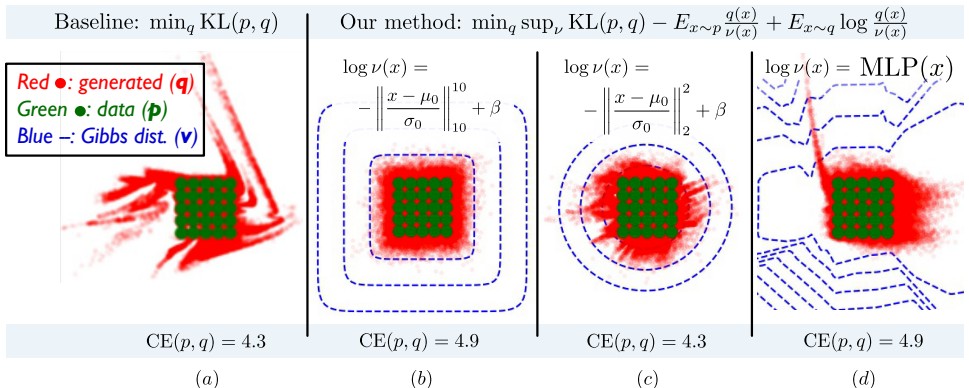

Figure 1: Different trained models for a 2D toy problem. The data distribution (samples shown as green dots) consists of a mixture of isotropic Gaussian distributions, arranged in a square grid. Four different trained models (generator samples shown as red dots) are shown: (a) result of training a model to optimize KL(p,q), (b-d) result of training models with our method (fine-tuning result of (a)), varying the form of the structured Gibbs distribution $\nu$. Blue lines show contours of learned structured Gibbs distributions $\nu$. In (b-c), the learned parameters of $\nu$ are $\mu_0$, $\sigma_0$, and $\beta$.

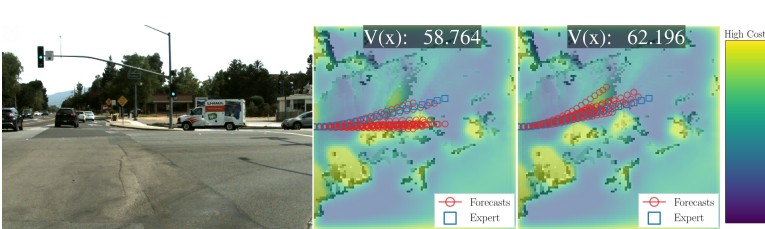

Figure 2: Our method applied to forecast ego-vehicle trajectories, showing input image and overhead views with the learned cost $-\log(\nu)$ (overlaid on LIDAR map). Middle shows samples from current $q$ (red), and true future path (cyan). Note that $\nu$ has learned to penalize regions with obstacles ($V(x) = \log \nu(x)$, *i.e.* higher $V(x)$ is lower traversal cost). Right: after incorporating the learned $\nu$, $q$ and its samples are shifted to avoid high-cost regions, corresponding to suppression of spurious modes.

autoregressive warp in two dimensions [29, 8, 18]. We observe that the generated samples effectively cover all the data modes, since a huge penalty is incurred if $q(x)$ is low for any data point $x \sim p$; however, the generator additionally places mass outside the support of the data distribution $p$, because shifting half of $q$'s mass onto the support of $p$ decreases $\text{KL}(p, q)$ by no more than $\log 2$; therefore, this objective effectively under-constrains $q$. We can resolve the ambiguity by adding the variationally-approximated $\text{KL}(q, p)$ term, which trains $\log \nu$ to approximate $\log p$ while penalizing the generation of samples where $\log \nu$ is low, as shown in Fig. 1b-d. By choosing different forms for $\log \nu$, we see that the ambiguity can be resolved in different ways: in Fig. 1b, $\log \nu$ is chosen to have contours roughly matching the support of $p$, whereas Fig. 1c-d show the cases where $\log \nu$ is represented as quadratic and as the output of a multi-layer perceptron, respectively. We see that matching the shape of $\log \nu$ to the shape of $p$ produces qualitatively good samples, whereas putting too little constraint on the shape of $\log \nu$ (as in Fig. 1d) yields qualitatively worse results. Estimated cross-entropy values are also shown for each model, which again demonstrates how $\text{KL}(p, q)$ is largely insensitive to the overall shape of the model $q$, as long as $q$ covers the modes of $p$.

Our method may also be viewed as incorporating a kind of f-GAN [26] optimizing a particular loss (Jeffrey divergence), using a (invertible) generator that admits exact PDF evaluation, and reparameterizing the discriminator in a certain way. The Fenchel-variational view espoused in [26] reveals the optimal discriminator $T$ to be a function of the odds ratio: $T^* = h(q/p)$ for some function $h$ determined by the particular $f$-divergence chosen. We observe that if $q$ can be evaluated analytically, then we might as well represent the discriminator in terms of $q$ and some function $\nu$, where $\nu$ directly approximates $p$. This change of representation allows us to take any known

regularities of $p$ and impose them on $\nu$; e.g., if $p$ is known to be translation-invariant, then we can safely impose translational invariance on $\nu$ without imposing any undue constraints on our model. Note that we cannot similarly impose $p$'s structure on $T$, which may be a complex function even if $p$ and $q$ are simple: for example, even if $p$ and $q$ are both bounded above, $T$ may be unbounded, since it is a function of the ratio of the two.

Fig. 2 illustrates how $\nu$ is structured for our application domain of vehicle trajectory forecasting: $\nu$ is represented as a sum of learned spatial rewards, which penalizes trajectories according to a learned function over spatial positions. Intuitively, this prevents trajectories from colliding with obstacles, while simultaneously learning the concept of an obstacle.

## 2 PUSHFORWARD REPRESENTATION AND IMPLEMENTATION

Optimizing Eq. (1) is straightforward except for one subtle point: we must be able to evaluate $q$ pointwise and differentiate the expression $\mathbb{E}_{x \sim q} \log(q(x)/\nu(x))$ with respect to the parameters of $q$. Inspired by prior work [29, 8, 18], we solve both these problems by representing $q$ as the *pushforward* of a simple distribution under an invertible warp (also known as a normalizing flow). Suppose $\mu$ is a distribution over a set $Z$ and $g : Z \to X$ is a function (the *generator*) with domain $Z$. Then we can define a measure on $X$ as the distribution of $g(z)$ sampling $z$ from $\mu$—this distribution, denoted here by $g|_\mu$, is referred to as the pushforward of $\mu$ under $g$.

Now suppose $g$ is parameterized by $\theta$ and differentiable in $x$ and $\theta$. By representing $q$ as $q_\theta = g_\theta|_\mu$, for some simple distribution $\mu$, we can move the derivative wrt. $\theta$ inside the expectation by exploiting the property $\mathbb{E}_{x \sim g_\theta|_\mu} f(x) = \mathbb{E}_{z \sim \mu} f(g_\theta(z))$ for all functions $f$:

$$\frac{\mathrm{d}}{\mathrm{d}\theta} \mathbb{E}_{x \sim q} \log \frac{q(x)}{\nu(x)} = \frac{\mathrm{d}}{\mathrm{d}\theta} \mathbb{E}_{x \sim q_\theta|_\mu} \log \frac{q(x)}{\nu(x)} = \mathbb{E}_{x \sim \mu} \frac{\mathrm{d}}{\mathrm{d}\theta} \log \frac{q_\theta(g_\theta(x))}{\nu(g_\theta(x))}. \tag{2}$$

This is well-known as the *reparameterization trick*; it allows us to obtain a low-variance, unbiased estimate of the parameter derivatives for learning with SGD. However, one problem remains: evaluating $q_\theta(x) = (g_\theta|_\mu)(x)$, which appears in both the first and last terms of (1). This is solved by assuming that $g_\theta$ is invertible: $\hat{z} := g_\theta^{-1}(\hat{x})$. Thus, we have an analytic formula for the pushforward density:

$$q_\theta(g_\theta(z)) = (g_\theta|_\mu)(g_\theta(z)) = \mu(z) \left|(\mathrm{d}g_\theta)_z\right|^{-1}, \tag{3}$$

where $|(\mathrm{d}g_\theta)_z|$ represents the determinant of the Jacobian of $g_\theta$ evaluated at the point $z$. This finally allows us to rewrite Eq. (1) in the following explicit form, after performing some simplifications:

$$-\max_\theta \inf_\phi \mathbb{E}_{\hat{x} \sim p} \log \frac{\mu(\hat{z})}{|(\mathrm{d}g_\theta)_{\hat{z}}|} + \frac{\mu(\hat{z})}{|(\mathrm{d}g_\theta)_{\hat{z}}|\nu_\phi(\hat{x})} + \mathbb{E}_{z \sim \mu} \log \frac{\mu(z)}{|(\mathrm{d}g_\theta)_z|\nu_\phi(g_\theta(z))}. \tag{4}$$

A pseudocode summary of our method is given in Algorithm 1. $X$ and $Z$ denote batches of observed and latent samples, respectively, while subscripts $D$ and $G$ denote either data or generated samples. A few implementation issues are noted here. Applying the method to a particular problem requires the implementation of the invertible generator $G(\cdot; \theta)$, the structured Gibbs energy $\log \nu(\cdot; \phi)$, and the base distribution $\mu$. In order to avoid numerical issues when $q$ is not absolutely continuous wrt. $\nu$ (i.e., when $\nu$ is 0 but $q$ is not), we reparameterize $\nu$ as $\nu \leftarrow \alpha q + \nu$, where $\alpha$ is a small number. Given this assumption, the quantity $q(x)/(\alpha q(x) + \nu(x))$ can be rewritten in terms of the sigmoid $\sigma$. Optimization proceeds by alternating between minimizing the loss (1) in the generator parameters $\theta$ and maximizing it in the energy parameters $\phi$. However, as noted in Alg. 1, we may alternatively minimize a different objective for the generator: namely, $-\mathbb{E}_{x \sim p} \log q(x) + \mathbb{E}_{x \sim q} \log(q(x)/\nu(x))$. The rationale for the alternative objective is that, as noted in Sec. 3.1, the inner minimization may be viewed as fitting $\nu$ to $p$—in which case, we may approximate $\mathrm{KL}(q, p)$ as $\mathbb{E}_{x \sim q} \log(q(x)/\nu(x))$. The alternate generator objective was used for the toy experiment in Fig. 1, whereas the original objective was used for the trajectory forecasting experiments.

## 3 DERIVATIONS AND INTERPRETATIONS

Equation (1) can be derived via a variational lower bound derived from Fenchel conjugacy, using a technique similar to [25, 26]. Our approach of pairing this convex conjugate with the pushforward

---

**Algorithm 1** Pseudocode for C3PO implementation

---

**Require:** $X_D$: a batch of training data, $Z_G$: batch of generator noise samples from $\mu$
1:   $Z_D$, $\det dg^{-1}_{X_D} \leftarrow G^{-1}(X_D; \theta)$   $\triangleright$ $G^{-1}(x)$ returns $g^{-1}_\theta(x)$ and log det. of Jacobian of $g^{-1}_\theta$ at $x$
2:   $X_G$, $\det dg_{Z_G} \leftarrow G(Z_G; \theta)$          $\triangleright$ $G(z)$ returns $g_\theta(z)$ and log det. of Jacobian of $g_\theta$ at $z$
3:   $\log q(X_D) \leftarrow \log \mu(Z_D) + \log |\det dg^{-1}_{X_D}|$          $\triangleright$ Generator PDF at data samples
4:   $\log q(X_G) \leftarrow \log \mu(Z_G) + \log |\det dg_{Z_G}|$          $\triangleright$ Generator PDF at generator samples
5:   $\log q/\nu(X_D) \leftarrow \log \alpha + \log q(X_D) - \log \nu(X_D; \phi)$
6:   $\log q/\nu(X_G) \leftarrow \log \alpha + \log q(X_G) - \log \nu(X_G; \phi)$
7:   $L \leftarrow \text{BatchMean}(-\log q(X_D) - \alpha^{-1}\sigma(\log q/\nu(X_D)) + \log \sigma(\log q/\nu(X_G)) - \log \alpha)$
8:   **for** $i \leftarrow 1 \ldots N$ **do**
9:      $\theta \leftarrow \theta - \beta \nabla_\theta L$    $\triangleright$ Alternative generator loss: $L := -\log q(X_D) + \log \sigma(\log q/\nu(X_G))$
10:     **for** $j \leftarrow 1 \ldots M$ **do**
11:        $\phi \leftarrow \phi + \beta \nabla_\phi L$

---

direct density estimation motivates our method's name: Convex Conjugate Coupled Pushforward Optimization (C3PO). Observe that the Jeffrey divergence can be written as $\min_q -\mathbb{E}_{\hat{x} \sim p} \log q(\hat{x}) + \mathbb{E}_{x \sim q} \log q(x) - \mathbb{E}_{x \sim q} \log p(x)$. Since $q$ can be sampled, evaluated, and differentiated (Sec. 2), but $p$ can only be sampled, our goal in this section is to convert $-\mathbb{E}_{x \sim q} \log p(x)$ to something expressable in terms of expectations wrt. $p$ and $q$. This is achieved by applying the Fenchel-Young inequality to the function $f(p) := -\log p$, which yields $-\log p \geq \sup_{\lambda < 0} \lambda p - (-1 - \log(-\lambda))$. Substituting this inequality in place of $-\log p(x)$ yields the following lower bound of $-\mathbb{E}_{x \sim q} \log p(x)$:

$$\int_X q(x)(\sup_{\lambda < 0} \lambda p(x) + \log(-\lambda) + 1)\, \mathrm{d}x = 1 + \sup_{\lambda < 0} \int_X q(x)\left(\lambda(x)p(x) + \log(-\lambda(x))\right)\, \mathrm{d}x, \quad (5)$$

where the sup on the right-hand side is taken over all functions $\lambda : X \to \mathbb{R}^-$ mapping the domain to a negative scalar. Observing that $\int q(x)\lambda(x)p(x)\, \mathrm{d}x$ can be written either as $\mathbb{E}_{x \sim q} \lambda(x)p(x)$ or $\mathbb{E}_{\hat{x} \sim p} \lambda(\hat{x})q(\hat{x})$, and making the substitution $\nu = -1/\lambda$, we have the equivalent bound

$$-\mathbb{E}_{x \sim q} \log p(x) \geq 1 + \sup_{\nu > 0} \mathbb{E}_{\hat{x} \sim p} - \frac{q(\hat{x})}{\nu(\hat{x})} - \mathbb{E}_{x \sim q} \log \nu(x). \quad (6)$$

Substituting this the Jeffrey divergence yields Eq. (1).

### 3.1 INTERPRETATION AS LEARNING A GIBBS DISTRIBUTION

Although we have so far viewed our goal as primarily learning $q$, we now observe that our method can also be interpreted primarily as a way of learning $\nu$ as a Gibbs distribution approximating $p$. Learning a Gibbs distribution, or a more general *energy-based model* [19], is a very general and effective way to impose strong domain-specific regularization on probability distributions over high-dimensional data. Generally, this is achieved by structuring the energy function $V_\phi := \log \nu_\phi$ to assign similar energies to similar examples; for example, a convolutional neural network might constitute a good energy for image generation, since the structure of a CNN encodes some degree of translational invariance. Unfortunately, inference and learning with high-dimensional Gibbs distributions is difficult.

Our method can be viewed as a way to train a Gibbs distribution that circumvents some of these difficulties. In this view, the inner optimization in Eq. (1) is interpreted as minimizing a weighted divergence between $p$ and $\nu$; the weights are exactly $q$. Specifically, we observe that the inner optimization in Eq. (1) minimizes the following weighted Itakura-Saito divergence [16, 9]:

$$\min_\phi \int q(x) \left( \frac{p(x)}{\nu_\phi(x)} - \log \frac{p(x)}{\nu_\phi(x)} - 1 \right) \mathrm{d}x \quad (7)$$

It would seem reasonable to choose the weights $q$ as $q = p$; since $p$ cannot be evaluated directly, alternating optimization is a sensible alternative. Minimizing Eq. (7) intuitively gives us the best Gibbs approximation of $p$ over the support of $q$.

Intuition for the learning rule of $\nu$ can be obtained by computing the *functional gradient* of Eq. (1) with respect to $\nu$ (i.e., differentiating with respect to $\nu(x)$, $\forall x$. The functional gradient $\delta/\delta\nu$ expresses

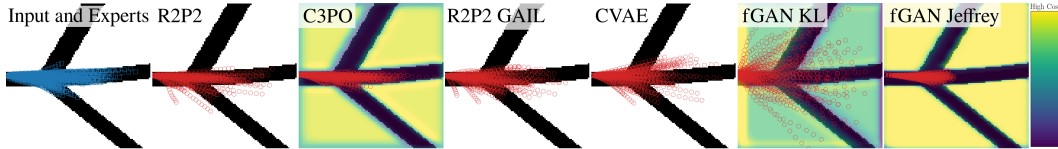

Figure 3: Comparison of methods on a test scene from the BEVWORLD1K dataset. Left column: The input BEV map (roads in black), accompanied by 100 demonstrations of possible behavior (in blue). Each method is visualized with 60 of each samples. For the methods that learn a cost map, the learned cost map is blended with the input BEV feature map.

Table 1: Comparison of methods in two datasets. Left: Single BEVWorld Scene (identical train and test), 300 experts, 1800 policy samples. Right: BEVWorld 100 training scenes, 1000 test scenes, 100 experts/scene, 12 policy samples/scene. Means and their standard errors are reported. Bold indicates the best performing method among methods with nondegenerate $-H(p, q)$ (*i.e.* fGAN Jeffrey is degenerate). CVAE's $-H(p, q)$ is estimated via MC sampling 300 times per scene, see [37].

| | BEVWORLD 1 | | | BEVWORLD1K | | |
|---|---|---|---|---|---|---|
| Method | $-H(p,q)$ | $-H(q, p_{\text{KDE}})$ | Road % | $-H(p,q)$ | $-H(q, p_{\text{KDE}})$ | Road % |
| R2P2 | $\mathbf{83.7} \pm 0.2$ | $-46.6 \pm 0.5$ | 0.988 | $\mathbf{96.0} \pm 0.03$ | $-61.6 \pm 0.6$ | 0.922 |
| C3PO (ours) | $82.6 \pm 0.3$ | $\mathbf{-44.8} \pm 0.4$ | $\mathbf{1.000}$ | $92.7 \pm 0.1$ | $\mathbf{-48.7} \pm 0.3$ | $\mathbf{0.989}$ |
| R2P2 GAIL | $64.6 \pm 2.0$ | $-55.6 \pm 1.0$ | 0.952 | $62.0 \pm 0.6$ | $-74.6 \pm 0.8$ | 0.886 |
| CVAE* | $18.6 \pm 3.7$ | $-45.3 \pm 1.7$ | 0.990 | $12.5 \pm 0.3$ | $-71.3 \pm 0.8$ | 0.865 |
| fGAN KL | $16.6 \pm 0.6$ | $-294.4 \pm 22$ | 0.568 | $18.6 \pm 0.03$ | $-303 \pm 2.6$ | 0.698 |
| fGAN Reverse KL | $-17.3 \pm 2.9$ | $-171 \pm 8.5$ | 0.706 | $-11.6 \pm 0.2$ | $-235 \pm 1.8$ | 0.709 |
| fGAN Jeffrey | $-7e4 \pm 6e3$ | $-42.0 \pm 0.05$ | 1.000 | $-5e3 \pm 39$ | $-46.1 \pm 0.08$ | 0.970 |

the direction in which $\nu$ should be moved at each point $x$ in order to optimally decrease the objective. Observing that $E_{x \sim p} q(x)/\nu(x) = E_{x \sim q} p(x)/\nu(x)$, we obtain the following for the functional gradient of the objective wrt. $\log \nu$: $\delta C / \delta \log \nu(x) = q(x) \left( p(x)/\nu(x) - 1 \right)$. We therefore see that minimizing Eq. (1) in $\nu$ raises or lowers $\log \nu$ at each point $x$ according to whether it exceeds $p(x)$, with a learning rate given by $q(x)$, and a unique fixed point (assuming $q > 0$) of $p = \nu$.

# 4 EXPERIMENTS

We conducted experiments on four datasets of vehicle trajectories, where the objective is to forecast a distribution over the vehicle's future locations $x \in \mathbb{R}^{20 \times 2}$ given contextual information about each scene. The contextual information includes 2 seconds of the vehicle's previous position, as well as a Bird's Eye View (BEV) map of visual scene features. Our experiments are designed to quantify two key aspects of generative modeling: the learned model's likelihood of held-out test data (*i.e.* the negative forward cross-entropy $-H(p, q)$), and the quality of samples from the learned model (*i.e.* the negative reverse cross-entropy $-H(q, p)$). Our hypotheses are: *1)* C3PO *will achieve superior sample-quality performance to other methods 2)* C3PO *will learn a high-quality q, partially due to its ability to perform direct density evaluation and optimization of q 3)* C3PO *will learn an interpretable $\nu$ that penalizes bad samples.*

Two of the four datasets are synthetic (BEVWORLD 1 and BEVWORLD1K), with known roads, enabling us to construct samples from a reasonable $p$ distribution, approximate $p$ via KDE, and measure the reverse cross-entropy $H(q, p_{\text{KDE}})$. We also calculate the percentage of trajectory points on the road as another measure of sample quality. Details of how we generate BEVWORLD are in the supplement. We also experiment with two real-world datasets: the KITTI dataset, and the CALIFORECASTING dataset [32].

## 4.1 IMPLEMENTATION AND BASELINES

Given our setting is that of forecasting future vehicle trajectories, we leverage ideas from Inverse Reinforcement Learning [24, 43, 41], to structure our Gibbs energy $V$ as a spatial *cost map*: where

Table 2: Comparison of methods in two real-world datasets: KITTI and CALIFORECASTING. Means and their standard errors are reported. Bold indicates the best performing method among methods with nondegenerate $-H(p,q)$ (*i.e.* fGAN Jeffrey is degenerate).

| | KITTI | | CALIFORECASTING | |
|---|---|---|---|---|
| Method | $-H(p,q)$ | $V_\phi^{\text{KITTI}}(q)$ | $-H(p,q)$ | $V_\phi^{\text{CALIF}}(q)$ |
| R2P2 | $\mathbf{63.7} \pm 0.8$ | $-744 \pm 20$ | $\mathbf{74.1} \pm 0.38$ | $-6.50 \pm 3.9$ |
| C3PO (ours) | $61.5 \pm 0.7$ | $-\mathbf{457} \pm 13$ | $73.5 \pm 0.4$ | $\mathbf{57.3} \pm 1.4$ |
| R2P2 GAIL | $54.9 \pm 0.7$ | $-693 \pm 17$ | $46.9 \pm 0.3$ | $-61.1 \pm 6.1$ |
| CVAE* | $9.22 \pm 0.9$ | $-555 \pm 9.9$ | $10.1 \pm 0.9$ | $48.3 \pm 1.5$ |
| fGAN KL | $32.9 \pm 1.3$ | $-693 \pm 10$ | $9.55 \pm 0.02$ | $-568 \pm 20$ |
| fGAN Reverse KL | $12.8 \pm 0.08$ | $-1362 \pm 33$ | $-89.7 \pm 3.1$ | $21.8 \pm 2.6$ |
| fGAN Jeffrey | $-2e4 \pm 2e3$ | $-195 \pm 4.0$ | $-2e4 \pm 7e2$ | $69.5 \pm 0.1$ |

$\log \nu_\phi = V_\phi(x) = \sum_{t=1}^{T} R_\phi(x_{t0}, x_{t1}; \text{BEV})$, and $R_\phi(a, b; \text{BEV})$ is the output of a CNN that can be interpolated at 2d positions of the form $(a, b)$. This structure enables $\nu$ to penalize trajectories that travel to locations it perceives to be bad, *e.g.* locations with obstacles, or locations far from a perceived road.

We compare our method, C3PO, to several state-of-the-art approaches in imitation learning and generative modeling: Generative Adversarial Imitation Learning (GAIL) [12], f-GAN [26], the CVAE method of DESIRE [20], and R2P2[32]. In each baseline, we use architectures as similar to our own method as possible: the policy (generator) architecture of GAIL and the generator architecture of f-GAN are identical to the generator architecture of our own approach. The same architecture used for $V_\phi$ in our method was also used for the discriminators in all baselines.

Our implementation of the $q$ architecture is based on R2P2 [31]. One key difference between our method, C3PO, and R2P2, is that R2P2 does not have an adversarial component; its main focus is learning the forward KL term, the first component of C3PO's objective function. R2P2 starts from a similar objective: the symmetric sum of cross-entropies. However, it relies on a cruder approximation of $H(q, p)$, because it learns an approximating $\tilde{p}$ for $H(q, \tilde{p})$ offline, with no interaction from $q$.

## 4.2 SYNTHETIC EXPERIMENTS

BEVWORLD 1contains a single scene with 300 samples from $p$. This setting is *unconditional*: the contextual information provided is identical, and not directly useful for modeling $p$. This setting also provides a fairer comparison to fGAN methods, which left the extension of fGAN to contextual settings as future work [26]. Table 1 shows the results of BEVWORLD 1 experiments. We observe that R2P2 and C3PO achieve the best $-H(p,q)$ scores, with C3PO outperforming all nondegenerate methods in its sample quality. This evidence supports hypotheses 1) and 2): C3PO achieves superior sample quality and high-quality data density. We also observed C3PO to be very stable throughout training in terms of $H(p,q)$: it was as easy to train it as R2P2 in all experiments.

We also observe that while fGAN KL indirectly learns a $q$ with some support for $p$, its samples are quite poor. This matches expectations, as the forward KL divergence fails to impose much penalty on sample quality [14]. fGAN Jeffrey is the fGAN method most similar to our approach because it uses the Jeffrey divergence. We observe it to suffer mode collapse in all of our experiments, a similar result to [26], in which fGAN Jeffrey had the worse test-set likelihood.

Next, we consider BEVWORLD1K, in which 100 training scenes, each with 100 samples from $p$, are used to learn *conditional* generative models. There are 1000 scenes in the test set. Table 1 shows the result of these experiments. We observe results similar to BEVWORLD1K, again supporting hypotheses 1) and 2). C3PO can learn in the conditional setting to produce high-quality samples from a distribution with good support of $p$. We display example results in Fig. 3, and observe several methods, including C3PO , learn a cost-map representation that penalizes samples not on the road. This evidence of the learned intuitive perceptual measurement of $\nu$ supports hypothesis 3).

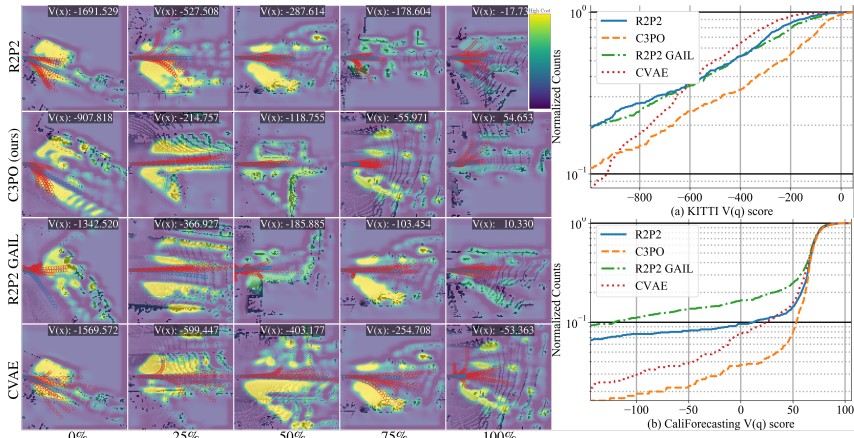

Figure 4: **Left:** Comparison of methods under the learned $V_\phi^{\text{KITTI}}(q)$ criterion. Each row corresponds to a method, and each column corresponds to the method's result on the item at a specific level of performance, from worst (left) to best (right) of results on 100 scenes. Each image is composed of the learned $V_\phi^{\text{KITTI}}(q)$ blended with the input BEV features, sample trajectories from each method (red), and the true future (blue). The learned $V$ often penalizes samples that go off of the road or into obstacles inferred from the features. C3PO usually produces the best samples under this metric. **Right:** Evaluation of sample quality on test data from the KITTI dataset (a) and CALIFORECASTING dataset (b). Each approach generated 12 trajectory samples per scene, and the mean $V_\phi$ score was calculated for each scene. The results are displayed as a normalized cumulative histogram, where the $y$-value at $x$ is the percentage of scenes that received $V \leq x$. At almost every given $V$, C3PO is likelier to have more samples above the given $V$ than other methods.

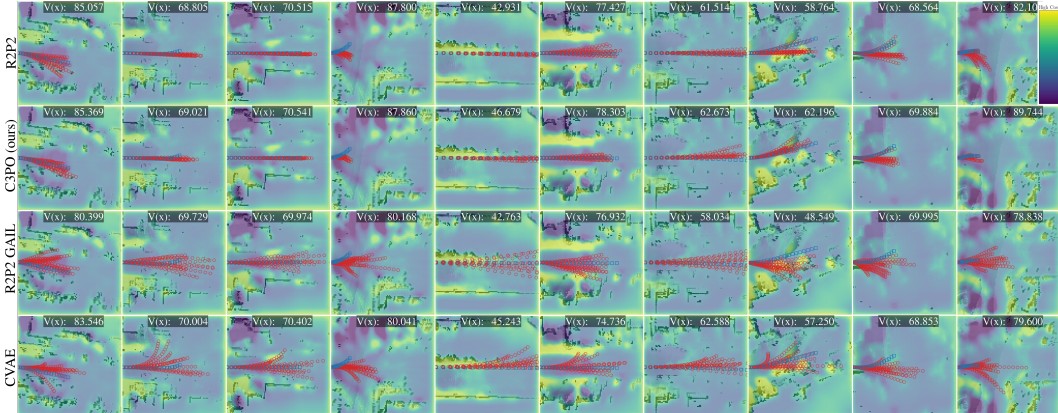

Figure 5: Comparison of methods on a set of random scenes from the CALIFORECASTING test set. Scene indices were selected uniformly at random (once) from the possible test indices. Each image shows the learned $V_\phi^{\text{CALIF}}(q)$ blended with the input BEV features, 12 sample trajectories from each method (red), and the true future (blue). $V$ learns to penalize samples that go off of the road or into obstacles.

### 4.3 REAL-WORLD EXPERIMENTS

We now experiment with real-world data, in which each method is provided with noisy contextual information, and is evaluated by its ability to produce a generative model and high-quality samples. Evaluation of sample quality is more difficult in this scenario: we have only one sample of $p$ in each scene, precluding construction of $p_{\text{KDE}}$, and there are no labels of physical roads, precluding computation of on-road statistics. Fortunately, evaluation of $-H(p, q)$ is still possible.

After observing the learned $V_\phi$ of C3PO on both synthetic and real data, we found $V_\phi$ to be generally interpretable and intuitively good: it assigns low cost to roads it learns to perceive and it assigns high cost to obstacles it learns to perceive. We therefore employed our learned model $V_\phi$ to quantitatively evaluate samples from all methods. Fig. 4 illustrates results from several approaches visualized with $V$ and point cloud features from the BEV, at various quantiles of each method's performance under $V$. Fig. 5 (left) illustrates each method on a set of randomly sampled scenes. In both figures, $V$ perceives and assigns penalty to regions around obstacles in the point cloud data. Note that this energy is learned implicitly, because demonstrations from $p$ avoid obstacles. These results provide further support for hypothesis 3).

Quantitative results are shown in Table 2. We find that results are, overall, similar to results on the other datasets. C3PO produces a high-performing $q$ in terms of both its likelihood, $-H(p, q)$, as well as the quality of its samples, $V$. Additionally, we show evaluation of $V$ for the top methods in Fig. 4 (right). Together, these results further strengthen evidence in support of all of our hypotheses.

## 5 RELATED WORK

The initial motivation for our work was the apparent dichotomy between sample quality and mode coverage in existing deep generative models; this phenomenon has been noted and quantified in work such as [40, 23, 4, 13, 14]. Our work synthesizes several techniques from prior work to address these issues, including the Fenchel-variational principle from work such as [25, 26], the well-known reparameterization trick [17, 35], and analytic pushforward-based density estimators such as normalizing flows [29, 18], RealNVP [8], and related models [22, 10, 27, 38, 30]. Comparatively little work has explored combining these methods, with some exceptions. Combining a RealNVP [8] density estimator with a GAN objective was considered in [11]; however, this is susceptible to the problems inherent with GANs mentioned in the introduction. A pushforward-based density estimator was employed in conjunction with variational inference to optimize the classical Bayesian evidence lower bound in [18], but this cross-entropy objective suffers from the previously-mentioned problems with optimizing $KL(p \parallel q)$ alone, as do all methods based on this objective, including [8, 10, 27, 38].

Our work is also comparable to the extensive literature on learning deep graphical models, including variants of Boltzmann machines [1, 34] and various proposals for learning deep CRFs and structured energy-based models [19] based on inference techniques including convex optimization [28, 7, 2, 36], various forms of unrolled inference [33, 42, 39], pseudolikelihood [21], and continuous relaxation [5], among other techniques [13]. Viewed as a way to learn a deep graphical model, the most important distinguishing factor of our work is the fact that our method does not need to perform inference directly, as discussed in Sec. (3.1). To the extent that $q$ is viewed as (indirectly) performing inference, our method and other deep generative models bears some similarity to the wake-sleep algorithm, which also alternates between optimizing a model distribution and an "inference" distribution using complementary divergences [13]. Score-matching [15] also learns a Gibbs distribution without inference; however, unlike our method, score-matching does not learn an inference distribution.

## 6 CONCLUSION

We have demonstrated how GANs may be effectively regularized by reparameterizing the discriminator to be a function of the model density and a structured Gibbs density, showing how this may be achieved by optimizing a Jeffrey divergence loss, assuming the generator is invertible, and applying a variational bound based on Fenchel conjugacy. Applied to a trajectory forecasting problem, we observed superior mode coverage and qualitative results compared to traditional GANs. We hope to soon demonstrate the general applicability of our approach by applying it to the task of image generation as well.

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
