# OpenReview forum: "Learning Gibbs-regularized GANs with variational discriminator reparameterization"
_ICLR.cc/2019/Conference_

### Official Review · AnonReviewer2 · 2018-11-02

**Rating:** 4
**Confidence:** 4

**Review:**

This paper combines a number of ideas to train generative models with (deep) structured constraints. The general idea is similar to Flow-GAN, which learns a normalizing flow-based generator by optimizing the negative loglikelihood with an augmented GAN loss. However, It’s difficult to impose prior structure information in the GAN framework. To address this problem, the authors proposed to minimize a so-called Gibbs-regularized variational bound of Jeffery divergence, which is the summation of KL and reverse KL divergence. The authors provide some justification that the Jeffery divergence works by yielding good mass-covering and mode-seeing properties.

It appears that the parameterization and adaptation of v throughout optimization is the key contribution of this work --- the technical details are not clear from the paper.

1.    Typo in the training objective (Eq .1):  the second (or the first) "sup" should be removed?

2.    Section 2.3 is very confusing. Particularly, how is the parameter \phi introduced? What’s the detailed update of \phi?
- "We now observe that our methods can also be interpreted as a way of learning v as a Gibbs distribution approximating p." If v_\phi(x) is a distribution, what’s the parametric form of v?
- "Generally, this is achieved by structuring the energy function V_\phi:=\log v_\phi." It seems that V_\phi(x) is a scalar-valued function that represents the negative energy of the distribution v_\phi(x), however, why the distribution is self-normalized? Specifically, why \int \exp(V_\phi) dx = 1? Otherwise, how the authors deal with the partition function \int \exp(V_\phi(x)).
- It is unclear to me why the inner loop optimization is connected with Itakura-Saito divergence minimization? The authors may consider including the detailed proofs?

3.    With the given description, the proposed algorithm is not easy to follow and implement by the reader. The paper would benefit from an Algorithm box with pseudocode.

If the authors can fully address the concerns above, I will consider changing the scores.

Other comments:
1. The empirical results are fairly weak. Similar datasets are used, the authors may consider evaluating their approach on various different tasks.

2. Duplicate citations – R2P2 [35] [36]

3. Other related papers:
 - Belanger et al., End-to-End Learning for Structured Prediction Energy Networks, ICML 17
- Tu et al., Learning Approximate Inference Networks for Structured Prediction, ICLR 18

---

> ### Author Response · Authors · 2018-11-27
> **Algorithm box added and paper revised for clarity**
>
> Thank you for your comments.  Please note that we have revised
> the manuscript to address them.
>
> 1. typo
> =======
>
> Yes, the extra sup is a typo.  Thanks for pointing it out.
>
> 2. Section 2.3 is confusing
> ===========================
>
> We apologize for the confusion.  We have tweaked this section in the revised manuscript (which is now Section 3.1) and have added an algorithm box for clarity.
>
> V_\phi(x) = log \nu_\phi(x) can be any arbitrary function of x with parameters \phi, although in order to gain the benefits of regularization, V_\phi should be structured appropriately for the domain.  For example, if x is an image, then V_\phi(x) could be a CNN with convolution weights \phi.
>
> We emphasize that \exp V_\phi(x) does not need to be explicitly normalized over x.  In theory, the V_\phi that maximizes the objective (the equation now called Eq. 1 in the latest revision) is automatically normalized, because the optimal value of \exp V_\phi is p (the data distribution).  Since p is normalized, the optimal \exp V_\phi must also be normalized.
>
> Note that in practice, \exp V_\phi may not be normalized due to incomplete optimization; however, optimizing it still has the desired effect of pushing up V_\phi where the data is present and pushing it down elsewhere.  This seems to be sufficient to make the method work in practice.
>
> How is the inner loop connected to Itakura-Saito divergence minimization?
> =========================================================================
>
> This is an important point that we have clarified in our updated draft. Eq. 7 (in the latest paper revision) is equivalent to the inner sup_\nu in Eq. 1, in the sense that the argmin_\nu of Eq. 7 is equal to the argmax_\nu of Eq. 1.  The only difference is that the sign has been flipped (converting it to a min) and a constant factor of -1-E_{x ~ q} log p(x) has been added.  The latter factor does not depend on the parameters \phi of the energy \nu_\phi and hence does not affect optimization over \phi.  Since Eq. 7 is equal to the Itakura-Saito divergence between p and \nu, the maximizer of Eq. 1 in \nu is equal to the minimizer of the Itakura-Saito divergence between p and \nu in \nu.
>
> One other subtle issue worth mentioning is our particular definition of the Itakura-Saito divergence, which is parameterized by a "weighting function" q.  This is a natural choice for two reasons: first, it allows us to express the divergence as an expectation (with respect to q), thus allowing us to optimize it via SGD; second, it ensures the divergence is bounded if there is a constant offset in the integrand.  Intuitively, a constant offset may occur since \nu is not assumed to be normalized.
>
> The Itakura-Saito divergence and its weighted variant may both be  derived as Bregman divergences.  We would be happy to provide more details in an appendix if desired.
>
> 3. Algorithm box?
> =================
>
> Thank you for this suggestion.  We have added an algorithm box, which we believe has significantly enhanced the clarity of the presentation and demonstrates that the method is actually fairly simple to implement in practice.

---

### Official Review · AnonReviewer1 · 2018-11-02
**Decent idea but need more motivating experiments.**

**Rating:** 5
**Confidence:** 5

**Review:**

The paper proposes to reparameterize the discriminator to be an explicit function of two densities so that one could inject domain specific knowledge easily. As the authors say, that one way to inject domain specific information is by learning an energy function. Making use of this intuition, authors proposed to  regularize the discriminator in
(GANs) framework by leveraging structured Gibbs distributions.

I found the introduction a bit hard to read. Otherwise paper is written in a readable way.

Something which I like about this paper, is authors use the proposed method for actual RL problems as compared to just image generation. I think this is important as well as interesting.  As a community we should be moving towards evaluating generative models for the problems where we actually want to use generative models for.

Some questions:

- I'm not sure if the paper is really novel as the authors themselves point out that it corresponds to adding adversarial component in R2P2.
- I also did not find results very convincing. As I said, its important to evaluate on RL problems ONLY if it makes sense on toy problems first. Like in the paper, authors made a big claim about reducing spurious modes, but it has not been demonstrated any where per se. May be authors can construct a toy problem in which they can show that the spurious mode issue, and how the proposed method kills these spurious modes.  This also reminds me of the literature in Boltzmann machines and more recently in Variational Walkback [1]. This could also be cited, and could be interesting to authors.

[1] Variational Walkback, https://arxiv.org/abs/1711.02282. The authors in Variational walkback also make the assumption p == q.

What would make the paper stronger ?
- Constructing toy problems in order to illustrate the mode coverage and spurious modes issue would be interesting.

---

> ### Comment · AnonReviewer1 · 2018-11-26
> **No Reply. Original Rating.**
>
> The authors did not reply.  In this situation, I stand by my original review.

---

> > ### Author Response · Authors · 2018-11-27
> > **Introduction has been revised around a toy example**
> >
> > We apologize for the delayed response, although we note that the discussion period is still open, as far as we understand. We appreciate your insightful comments and have uploaded a revised manuscript taking them into account.  In particular, we have rewritten the introduction around results from an illustrative toy experiment (Fig. 1), which we believe significantly strengthens the motivation and readability of the work.
> >
> > What are "spurious modes"?
> > ==========================
> >
> > Please note that we used the term "spurious modes" as a somewhat imprecise shorthand and have deemphasized this language in the latest draft.  A more accurate description of the phenomenon we address is the inappropriate placement of model distribution mass outside the support of the data distribution, since the local maximum property of a mode is not necessary for the manifestation of the phenomenon. Fig. 1a shows a failure case of the baseline method where model mass is placed outside the support of the data distribution, but the model does not clearly exhibit multiple modes.
> >
> > The mental image of a spurious mode comes from the following illustrative example.  Consider minimizing KL(p,q) over q for fixed data PDF p.  Now consider the value of KL(p,q'), where q' = 0.5 * (p + \eta), assuming \eta is such that supp(p) \cap supp(\eta) = \emptyset. It is easy to show that KL(p,q') = log 2, and generalizing the argument shows that KL(p,q') = log N if q' is a mixture of N equally-likely components such that only one is equal to p, and the rest are disjoint from p.  Visualizing q' as a mixture of components mostly disjoint from p is what gives rise to the image of spurious modes.  It is also practically useful to think of KL(p,q') as penalizing q' only for the log of the number (~N) of spurious modes that it includes.
> >
> > We preferred to use related citations to provide evidence for the "spurious mode" phenomenon to save space, but we can include the technical argument above if those citations are considered insufficient.
> >
> > Are "spurious modes" real?  Does our method fix them?
> > =====================================================
> >
> > In addition to the arguments above and the toy example, which show the existence of the phenomenon and the ability of our method to fix it, we would also like to point out that the synthetic experiments (Table 1) clearly demonstrate that C3PO significantly reduces the number of model-generated examples falling outside the support of the data distribution.  We can measure this by calculating the percent of generated samples that go off-road, as samples from the training data never go off-road.  Table 1 shows that about 99% of C3PO's samples stay on-road while retaining a high log-likelihood value (indicating good mode coverage), whereas only 92% of the baseline model's samples stay on-road.  See Section 4.2 for details.
> >
> > How novel is our work?
> > ======================
> >
> > Although it is true that our technical innovation over R2P2 largely consists of applying Fenchel-variational inference to the KL(p,q) term and proposing to regularize the resulting model by imposing Gibbs structure on the variational function, we believe that to view our work only in these terms would be overly reductive, overlooking its potential impact.
> >
> > Our work bridges a gap between two camps in the generative modeling community: namely, deep generative models, which underestimate the importance of regularization; and classical energy-based probabilistic models, which have not yet realized how recent variational methods can help sidestep the obstacle of partition function estimation.  Our work bridges this gap by showing how one method can be seen as both a regularized deep generative model (Sections 1-2) and a novel way to learn an unnormalized graphical model without explicit partition function estimation, via iterative Itakura-Saito divergence minimization (Section 3.1).  We believe these insights are novel and will be interesting to a sizable fraction of the ICLR community.

---

### Official Review · AnonReviewer3 · 2018-11-02
**Regularization of GANs to remove spurious modes - but is this what is needed?**

**Rating:** 5
**Confidence:** 3

**Review:**

Summary: The paper tries to answer the problems of regularizing GANS. They reparametrize the discriminator to be an explicit function of two densities: the generator probability density function q and a structured Gibbs distribution v.

Comments:
1: This paper focuses on mode coverage problems, where spurious modes of learned model(q) not supported by target model(p) are pruned off.  It is not clear why this is a significant problem.  GAN trained models typically suffer from mode collapsing, requiring additional noise injection to support generation of diverse data.  This work seems to argue that the opposite is worth paying attention to, focusing on removal of modes.

2: The implementation of the architecture is similar to R2P2, except for the introduction of a new adversarial component. But according the evaluation in table 1 and table 2, we see that baseline model R2P2 performs better in -H(p,q) and for -H(q, pKDE) the value is near equal to their model.

3: They assume the generator is invertible, which enables the analytic evaluation of the q. But no supporting evidence or design architecture for the statement above is provided.

4: The explanation of imposing structure on the model distribution is not clear. In the introduction they first claim “we cannot impose structure directly on the joint distribution of a GAN’s outputs.” But after they claim “we submit that regularizing the structure of a GAN’s generator and discriminator is generally more difficult than imposing meaningful structure directly on the model distribution, which we will refer to as q. These two statements conflict because the model distribution is a joint distribution of GAN’s outputs.


6 Typos:
1)	in equation(1) we should minimize q for all the terms.
2)	in equation(1) first term is unrelated with v.
3)	in equation(1) the sup is for the last two terms.
4)	in equation(2) in RHS of equation the first term q_ |  should be q_ .

7: Writing could be improved.

8: In table 2 what’s the meaning of evaluation metric Road%

---

> ### Author Response · Authors · 2018-11-27
> **We address both spurious modes and mode loss.  Please see revised paper.**
>
> Thank you for your comments.  Please see the revised paper, which
> incorporates the points brought up by you and the other reviewers.
>
> 1. Is "mode removal" really necessary?
> ======================================
>
> Unfortunately, the old introduction may have caused some confusion.  We have rewritten the introduction and included a toy experiment to illustrate the main concepts.
>
> Our method attempts to prevent both the potential failure modes of deep generative models---specifically, mode collapse and "spurious modes."  We do this by starting from a baseline model (min_q KL(p,q), i.e., "max likelihood") that does not suffer from mode collapse, and then adding a loss component (KL(q,p)) that prevents the "spurious mode" problem of the baseline model.  However, it is also perfectly   valid to think of our method as starting with the loss that prevents spurious modes (KL(q,p)), but suffers from mode loss, and then adding  the component that prevents mode loss KL(p,q).  Either way, we argue  that preventing both failure modes is important.
>
> Please also see comments to AnonReviewer1 on "spurious modes."
>
> 2. Doesn't R2P2 perform just as well?
> =====================================
>
> Our method optimizes an objective that is essentially a trade-off between covering the support of the data as well as possible and not generating any samples outside the support of the data.  The results show that C3PO does a better job of managing this trade-off than R2P2: in Table 1, BEVWorld1K, C3PO decreases H(q,p) by 12.9 nats while suffering an increase of only 3.3 nats in H(p,q) compared to R2P2. Please note that this is a logarithmic scale: decreasing H(q,p) by 12.9 nats means that C3PO's samples are about e^12.9 = 400,000 times more likely under the data than R2P2's samples.  The BEVWorld1 scenario is more of a sanity check, as it includes only one train/test scene, while BEVWorld1K features 100 training scenes and 1000 test scenes.
>
> 3. Is invertibility a strong requirement?
> =========================================
>
> Our work builds on several recent papers that demonstrate the feasibility of using invertible generators to enable efficient evaluation of the generator PDF, including [8,18,29,A] (see latest paper revision and [A] below for citations).  Also, [31] notes that invertibility of the generator is a particularly natural assumption for the application of trajectory forecasting.  As noted in the paper, we borrow the generator architecture of [31] for our work.
>
> [A] Diederik P Kingma and Prafulla Dhariwal. "Glow: Generative Flow
> with Invertible 1x1 Convolutions." In: NIPS 2018.
>
> 4. Imposing structure on the model PDF
> ======================================
>
> We agree that this wording was unfortunate and have revised the introduction to improve overall clarity.  We meant to have q represent an abstract model PDF, which could either be implemented as the distribution of a generator's outputs or as a structured Gibbs distribution.  Our claim is that regularizing a 'q' represented as the distribution of a generator's outputs is more difficult than regularizing a 'q' represented as an unnormalized Gibbs distribution.
>
> This is illustrated in Fig. 1.  Fig. 1a shows the effect of poor regularization of the generator PDF combined with an inadequate loss: artifacts appear in the generated samples due to the peculiarities of the generator structure.  We remedy this by using variational inference to train an unnormalized Gibbs distribution \nu to mimic the data PDF p, and then penalizing the generator for generating samples where \nu is low.  It is much easier to control the "shape" of \nu than it is to control the shape of the generator's PDF, because the latter is determined by the peculiarities of the generator structure, whereas the shape of \nu is specified directly.  By controlling the shape of \nu, we can get different regularization effects on the generator.
>
> Please also see response to AnonReviewer1.
>
> 8. What’s the meaning of “Road %”
> =================================
>
> In the synthetic experiment, we generate a dataset consisting of simulated overhead road maps and corresponding paths.  All of the training paths stay on the roads.  “Road %” specifies the percent of generated paths that stay on the road.  Like KL(q,p), this is a measure of the method’s ability to generate samples that are likely under the true data distribution.  We will clarify this in the next version of the manuscript.

---

### Meta-Review · Area_Chair1 · 2018-12-14
**Intersting idea, but novelty is limited and experimental analysis could be extended.**

**Confidence:** 3
**Recommendation:** Reject

**Metareview:**

The paper proposes to define the GAN discriminator as an explicit function of a invertible generator density and a structured Gibbs distribution to tackle the problems of spurious modes and mode collapse. The resulting model is similar to R2P2, i.e. it can be seen as adding an adversarial component to R2P2, and shows competitive (but no better) performance. Reviewers agree, that these limits the novelty of the contribution, and that the paper would be improved by a more extensive empirical evaluation.